# Sequential Context Encoding for Duplicate Removal

**Lu Qi**[1]    **Shu Liu**[1,3]    **Jianping Shi**[2]    **Jiaya Jia**[1,3]
[1]The Chinese University of Hong Kong    [2]SenseTime Research    [3] YouTu Lab, Tencent
{luqi, sliu, leojia}@cse.cuhk.edu.hk    shijianping@sensetime.com

## Abstract

Duplicate removal is a critical step to accomplish a reasonable amount of pre-dictions in prevalent proposal-based object detection frameworks. Albeit simple and effective, most previous algorithms utilize a greedy process without making sufficient use of properties of input data. In this work, we design a new two-stage framework to effectively select the appropriate proposal candidate for each object. The first stage suppresses most of easy negative object proposals, while the second stage selects true positives in the reduced proposal set. These two stages share the same network structure, *i.e.*, an encoder and a decoder formed as recurrent neural networks (RNN) with global attention and context gate. The encoder scans pro-posal candidates in a sequential manner to capture the global context information, which is then fed to the decoder to extract optimal proposals. In our extensive experiments, the proposed method outperforms other alternatives by a large margin.

## 1    Introduction

Object detection is a fundamentally important task in computer vision and has been intensively studied. With convolutional neural networks (CNNs) [15], most high-performing object detection systems [15, 20, 16, 32, 7, 23, 19, 27] follow the proposal-base object detection framework, which first gathers a lot of object proposals and then conducts classification and regression to infer the label and location of objects in the given image. The final inevitable step is *duplicate removal* that eliminates highly overlapped detection results and only retains the most accurate bounding box for each object.

**State-of-the-Art:**  Most research on object detection focuses on the first two steps to generate accurate object proposals and corresponding class labels. In contrast, research of duplicate removal is left far behind. NMS [12], which iteratively selects proposals according to the prediction score and suppresses overlapped proposals, is still a popular and default solution. Soft-NMS [3] extends it by decreasing scores of highly-overlapped proposals instead of deleting them. Box voting [14, 26] improves NMS by grouping highly-overlapped proposals for generating new prediction. In [4], it shows that to learn the functionality of NMS automatically with a spatial memory is possible. Most recently, relation network [22] models the relation between object proposals with the same prediction class label.

**Motivation:**  Optimal duplicate removal is to choose the only correct proposal for each object. The difficulty is that during inference we actually do not know what is the object. In the detection network, we already obtain the feature of region of interest (RoI) for classification and regression. But this piece of information is seldom considered in the final duplicate removal when the score and location of each proposal candidate are available. It may be because the feature data is relatively heavy and people think it is already abstracted in the candidate scores. If this is true, using it again in the final step may cause information redundancy and waste computation.

So the first contribution of our method is to better utilize different kinds of information into duplicate estimation. We surprisingly found that it is very useful to improve candidate selection. The features

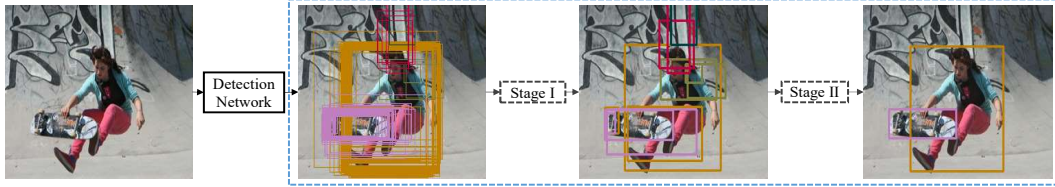

Figure 1: Illustration of our two-stage sequential context encoding framework.

form the global view of understanding objects rather than only considering a single category or independent proposals based on scores.

The second major contribution is the way to process candidate data. We take the large number of proposal candidates as a sequence data including its unique structure, and adopt recurrent neural networks (RNN) to extract vital information. It is based on our thoughtful design to generate prediction from a more global perspective in the entire image and make full use of the bi-directional hidden states.

Our final RNN-based duplicate removal model is therefore by nature different from previous solutions [3, 26, 22, 14, 21, 4]. It sequentially scans all object proposals to capture global context and makes final prediction based on the extra helpful information. Due to the enormous difference between proposal candidate and ground truth object numbers, our model is divided into two stages and performs in a way like boosting [13]. The first stage suppresses many easy negatives and the second performs finer selection. The two stages are with the same network structure, including encoder and decoder as RNNs, along with context gate and global attention modules.

Our method achieves consistent improvement on COCO [25] data in different object detection frameworks [23, 27, 8, 19]. The new way to utilize RNN for duplicate removal makes the solution robust, general and fundamentally different from other corresponding methods, which will be detailed more later in this paper. Our code and models are made publicly available.

## 2 Related Work

**Object Detection**    DPM [12] is representative before utilizing CNN, which considers sliding windows in image pyramids to detect objects. R-CNN [15] makes use of object proposals and CNN, and achieves remarkable improvement. SPPNet [20] and Fast R-CNN [16] yield faster speed by extracting global feature maps. Computation is shared by object proposals. Faster R-CNN [32] further enhances performance and speed by designing the region proposal network, which generates high-quality object proposals with neural networks. Other more recent methods [23, 19, 7, 27, 8, 27] improve object detection by modifying network structures.

Another line of research followed the single-stage pipeline. YOLO [31], SSD [28] and RentinaNet [24] regress objects directly based on a set of pre-defined anchor boxes, achieving faster inference speed. Although these frameworks differ in their operation aspects, the duplicate-removal step is needed by all of them to finally achieve decent performance.

**Duplicate Removal**    Duplicate removal is an indispensable step in object detection frameworks. The most widely used algorithm is non-maximum suppression (NMS). This simple method does not consider any context information and property of input data – many overlapped proposals are rejected directly. Soft-NMS [3] instead keeps all object proposals while decreasing their prediction scores based on overlap with the highest-score proposal. The limitation is that many proposals may still be kept in final prediction. Box voting [14, 26] makes use of information of other proposals by grouping highly-overlapped ones. With more information used, better localization quality can be achieved.

Desai *et al.*[10] explicitly encoded the class label of each candidate and their relation with respect to location. Final prediction was selected by optimizing the loss function considering all candidates and their relation. Class occurrence was considered to re-score object proposals in DPM [12] to slightly improve performance. More recently, GossipNet [21] processed a set of objects as a whole for duplicate removal with a relatively complex network structure with higher computation complexity. Spatial memory network [4] improved NMS by utilizing both semantic and location information. Relation network [22] models the relation between different proposals with the attention mechanism, taking both appearance and location of proposals into consideration.

Different from all these methods, we utilize an encode-decoder structure with RNN to capture and utilize the global context. With only simple operations, consistently better performance is achieved on all detection frameworks we experimented with.

**Sequence Model** RNN has been successfully applied to many sequence tasks like speech recognition [18], music generation [17], machine translation [2] and video activity recognition [11, 9, 1]. In neural machine translation (NMT), the concept of attention becomes common in training networks [29, 5, 30, 2, 33], allowing models to learn alignment between different modalities. In [2], parts of a source sentence were automatically searched that are relevant to predicting a target word. All source words were attended and only a subset of source words were considered at a time [29]. Intuitively, generation of content and functional words should rely much on the source and target context respectively. In [33], context gates dynamically control the ratios, at which source and target context contributes to the generation of target words.

## 3 Motivation

We first analyze the necessity and potential of duplicate removal. We take the three representative object detection systems as baselines, which include FPN [23], Mask R-CNN [19] and PANet [27] with DCN [8]. FPN can yield high-quality object detection results. Mask R-CNN is designed for instance segmentation, suitable for multi-task training. PANet with DCN achieves state-of-the-art performance on both instance segmentation and object detection tasks in recent challenges, which is a very strong baseline.

| Model | No Removal | NMS | Score (Oracle) | IoU (Oracle) |
|---|---|---|---|---|
| FPN [23] | 10.3 | 37.1 | 47.3 | 65.2 |
| Mask R-CNN [19] | 12.0 | 38.9 | 49.3 | 63.4 |
| PANet [27] with DCN [8] | 10.6 | 43.7 | 53.9 | 68.9 |

Table 1: Performance by modifying the duplicate removal step on COCO data [25].

In terms of the importance of duplicate removal, we explore performance drop for different detection methods without the final candidate selection, which simply set the final prediction as proposals when class labels and scores are higher than a threshold. As shown in the "No Removal" column of Table 1, three frameworks only achieve around 11 points in terms of mAP, with a decrease of more than 20 points. This experiment manifests the necessity of duplicate removal.

Then we evaluate the potential of improving final results when the duplicate removal step gets better. It is done by exploring the tight upper-bound of performance given ground-truth objects during testing. For each ground truth object, like NMS, we only select the proposal candidate with the largest score and meanwhile satisfying the overlap threshold. With these optimal choices, as shown in the "Score Oracle" column, the performance of all three baseline methods are much enhanced with 10+ points. This experiment shows there in fact is much room for improvement at the duplicate removal step.

Other than potential improvement, we also conduct experiments to evaluate the influence of inevitable proposal score errors during proposal generation. They inevitably influence duplicate removal since the scores are the most prominent indication of proposal quality and are utilized by methods like NMS, Soft-NMS and box voting to select proposals. In our experiments, we select the proposal candidates with the largest overlap with its corresponding ground truth. The results shown in the "IoU Oracle" column manifest that traditional NMS methods are likely to be influenced by the quality of prediction scores. Unlike NMS that only considers scores, our method has the ability of suppressing proposals with high prediction scores but low localization quality.

## 4 Our Approach

The key challenge for duplicate removal is the extreme imbalance of proposal candidate and ground truth object numbers. For example, a detection network can generate 1,000+ proposal candidates for each class compared with 10 or 20 ground-truth objects, making it hard for the network to capture the property of the entire image. To balance the positive and negative sample distributions, our framework cascades two stages to gradually remove duplicates, in a way analogous to boosting. This is because

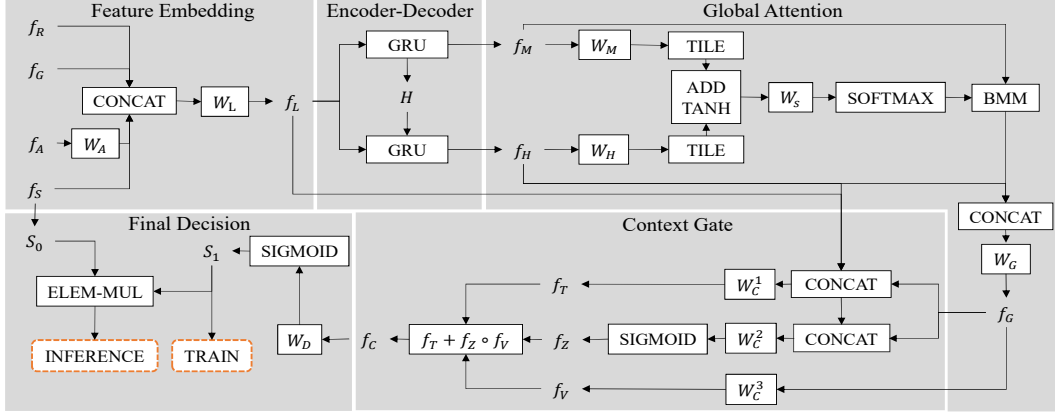

Figure 2: Details of our network components including feature embedding, encoder-decoder, global attention, context gate and final decision.

within any single image an overwhelming majority of proposal candidates are negative. As such, the cascade attempts to reject as many negatives as possible at the early stage [35].

Stage I suppresses easy negatives, which occupy a large portion of input object proposals in Fig. 1. In stage II, we focus on eliminating remaining hard negative proposals. These stages share the same network structure, including feature embedding, encoder, decoder, global attention, context gate, and final decision, for convenience. These components are deliberately designed and evaluated to help our model make comprehensive decision with multi-dimensional information.

Briefly speaking, we first transform primitive features extracted from object detection network for each proposal to low-grade features through feature embedding. Then the encoder RNN extracts middle-grade features to obtain the global context information of all proposals, stored in the final hidden state of the encoder. The decoder inherits the global-context hidden state and re-scans the proposal candidates to produce high-grade features. Global attention manages to seek the relation for each proposal candidate by combining the middle- and high-grade features. In case of missing lower-layer information at top of the network, the context gate is employed to selectively enhance it. The refined feature vector of each proposal helps determine whether the candidate should be kept or not. The overall network structure is showed in Fig. 2.

## 4.1 Feature Embedding

Features output from the object detection network are semantically informative. We extract appearance feature $f_A$, geometric feature $f_G$, and score feature $f_S$ for each proposal, where $f_S$ is a 1D prediction class score, $f_A$ is 1,024D feature from the last fully-connected ($fc$) layer in the proposal subnet in detection, and $f_G$ has 4D prediction coordinates. Given $f_G$ and $f_S$ abstract representation of $f_A$ in the detection network, 'smooth' operation for $f_A$ is needed before fusion of $f_G$, $f_S$, and $f_A$.

To this end, appearance feature $f_A$ is non-linearly transformed into $d_l$-D. Meanwhile, to maintain the scale-invariant representation for each bounding box, we denote $f_G$ as $\left(\log\left(\frac{x_1}{w} + 0.5\right), \log\left(\frac{y_1}{h} + 0.5\right), \log\left(\frac{x_2}{w} + 0.5\right), \log\left(\frac{y_2}{h} + 0.5\right)\right)$ where $(x_1, y_1, x_2, y_2)$ are the top-left and bottom-right coordinates of the proposal and $(w, h)$ are image width and height.

Intuitively, the closer proposal candidates are, the more similar their scores and appearance features are. To make our network better capture the quality information from the detection network, we rank proposal candidates in a descending order according to their class scores. Each proposal candidate is with a rank $\in [1, N]$ accordingly. The scalar rank is then embedded into a higher-dimensional feature $f_R$ using positional encoding [34], which computes cosine and sine functions with different wavelengths to guarantee the orthogonality for each rank. The feature dimension $d_r$ after embedding is typically 32.

To balance the importance of features, the geometric feature $f_G$ and score feature $f_S$ are both tiled to $d_r$ dimensions. Then transformed $f_A$, tiled $f_G$, tiled $f_S$ and $f_R$ are concatenated and then transformed

into smoother low-grade feature $f_L$ as

$$f_L = Max\{0, W_L \times Concat[Max(0, W_A f_A), f_S, f_R, f_G]\}. \tag{1}$$

## 4.2 Encoder-decoder Structure

It is hard for RNN to capture the appropriate information if the sequence data is in a random order. To alleviate this issue, we sort proposals in a descending order according to their class scores. So proposal candidates with higher class scores are fed to the encoder or decoder earlier. Moreover, each proposal has the context found in other proposals to encode global information. To make good use of it, we choose bi-directional gated recurrent units network (GRU) as our basic RNN model. Compared with LSTM, GRU is with fewer parameters and performs better on small data [6]. Its bi-direction helps our model capture global information from two orders.

For each stage, the encoder takes $f_L$ as input and outputs the middle-grade feature $f_M$. Different from zero initialization of the hidden state for encoder, the decoder receives the hidden state of encoder at the final time step with context information in proposals, basis for the decoder to re-scan $f_L$ to obtain the high-grade feature $f_H$. The size of hidden state in GRU is the same as input feature. Given the imbalance of class distributions, similar to traditional NMS and relation network [22], our method applies to each class independently.

## 4.3 Global Attention

Even though we pass the hidden state at the final time step of encoder to decoder, it is still hard for hidden state to embed all global information. As a remedy, we enable the decoder to access representation of each proposal in encoder, leading to better utilization of all proposals.

Since input data and structures of our encoder and decoder are identical except for their initialized hidden states, the output vectors tend to be similar, making it difficult for vanilla attention approaches [29] to capture their underlying relation. To address this issue, we apply a mechanism similar to Bahdanau attention [2] to first transforms the output of encoder and decoder into two different feature spaces, and then learn their relation. The detail is to calculate a set of attention weights $S_a$ for middle-grade feature $f_M$ as

$$S_a = \mu\{W_S \times Tanh[Tile(W_M \times f_M) + Tile(W_H \times f_H)]\}, \mu = Softmax, \tag{2}$$

where $f_M$ and $f_H$ are both linearly transformed and tiled. The *tile* operation is to get a new view of the feature with singleton dimensions expanded to the size of $f_M$ or $f_L$, such as tiling the vector from $N \times d_m$ to $N \times N \times d_m$ where N denotes the number of proposal candidates. By mapping $f_M$ and $f_H$ to different feature spaces, our attention could focus more on other proposal candidates.

Finally, we obtain the global feature $f_G$ by combining and smoothing $f_M$ and $f_H$ as

$$f_G = Max(0, W_G \times Concat[S_a \times f_M, f_H]). \tag{3}$$

## 4.4 Context Gate

In neural machine translation, generation of a target word depends on both source and target context. The source context has a direct impact on the adequacy of translation while target context affects fluency [33]. Similarly, in case of missing part of information, we design the context gate to combine the low-grade feature $f_L$, high-grade feature $f_H$ and global feature $f_G$. The benefit of context gate is twofold. First, like the residual module, it shortens the path from low to high layers. Second, it can dynamically control the ratio of contributions in low- and high-grade context.

We calculate gate feature $f_Z$ through

$$f_Z = \sigma[W_C^2 \times Concat(f_L, f_H, f_G)], \sigma = Sigmoid. \tag{4}$$

Then the source feature $f_V$ and target feature $f_T$ are obtained by

$$f_V = W_C^3 \times f_G, \ f_T = W_C^1 \times Concat(f_L, f_H), \tag{5}$$

where $f_Z$ is the combination of $f_L$, $f_H$ and $f_G$. $f_V$ is the linear transformation of $f_G$. $f_T$ is the linear transform of $f_L$ and $f_H$.

To control the amount of memory used, we only let $f_Z$ affect the source feature $f_S$, essentially like the reset gate in the GRU to decide what information to forget from the previous state before transferring information to the activation layer. The difference is that here the "reset" gate resets the source features rather than the previous state, *i.e.*,

$$f_C = \delta\left(f_T + f_Z \cdot f_V\right), \ \delta = \text{Tanh}, \tag{6}$$

where $\cdot$ means element-wise multiplication.

In the final decision, we obtain the score for each proposal candidate $s_1$ as

$$s_1 = \sigma\left(W_D \times f_C\right). \tag{7}$$

### 4.5 Training Strategy

The binary cross entropy (BCE) loss is used in our model for both stages. The loss is averaged over all detection boxes on all object categories. Different from that of [22], we use $L = -\log(1 - s_1)$ instead of $L = -log(1 - s_0 \cdot s_1)$, where $s_1$ denotes the output score of our model and $s_0$ denotes the prediction score of the proposal candidate from the detection network. Training with $s_0$ may prevent our model from making right prediction for proposal candidates mis-classified by detection network. Thus $s_0$ is not used in our training phase. $s_0 \cdot s_1$ is the final prediction score in inference to make use of information from both detection network and our model.

Our first stage takes the output of NMS as the ground-truth to learn and the second stage takes the output from stage I and learn to select the appropriate proposals according to the actual ground-truth object. Specifically, proposals kept by NMS are assigned positive labels in stage I. While in stage II, for each object, we first select proposals with intersection-over-union (IoU) higher than a threshold $\eta$. Then the proposal with highest score in this set are assigned positive label and others are negatives. By default, we use $\eta = 0.5$ for most of our experiments. Considering the COCO evaluation criterion $(\text{mAP@}0.5 - 0.95)$, we also extend multiple thresholds simultaneously [22], *i.e.*, $\eta \in [0.5, 0.6, 0.7, 0.8, 0.9]$. The classifier $W_D$ in Eq. 7 thus outputs multiple probabilities corresponding to different IoU thresholds, resulting in multiple binary classification heads. During inference, the multiple probabilities are simply averaged as a single output.

There are two ways to train our two-stage framework. The first is sequential to train stages I and II consecutively. The second method is to jointly update the weight of stage I during training stage II. Performance of our method in these two ways is comparable. We thus use sequential training generally.

## 5 Experiments

All experiments are performed on challenging COCO detection datasets with $80$ object categories [25]. 115k images are used for training [23, 22]. Ablation studies are conducted on the 5k validation images, following common practice. We also report the performance on *test-dev* subset for comparison with other methods. The default evaluation metric – AP averaged on IoU thresholds from 0.5 to 0.95 on COCO – is used.

As described in Section 3, we take FPN [23], Mask R-CNN [19] and PANet [27] with DCN [8] as the baselines to show the generality of our method. These baselines are implemented by us with comparable performance reported in respective papers.

For both stages in the framework, we adopt synchronized SGD as the optimizer and train our model on a Titan X Maxwell GPU, with weight decay $0.0001$ and momentum $0.9$. The learning rate is $0.01$ in the first ten epochs and $0.001$ in the last two epochs. $d_l$ and $d_m$ are by default 128 and 256.

In each training iteration, our network is with $0.45$ million parameters. This overhead is small, about $1\%$ in terms of both model size and computation compared to $43.07$ million parameters in FPN with ResNet-50. It takes about $0.019$s for the whole inference process with a single GPU, compared with $0.07$s by FPN with ResNet-50. Also, our computation cost is consistent even on larger backbone networks for object detection.

### 5.1 Stage I Performance

In stage I, we take the proposal candidates satisfying $s_0 \geq 0.01$ as input. The ground-truth labels are generated by NMS with IoU threshold $0.6$, which produce decent results with NMS. To reduce imbalance between positive and negative samples, the weight of positive samples in our BCE loss is set to $4$.

| Model | NMS | RNN | + Global Attention | + Context Gate | + Both |
|---|---|---|---|---|---|
| FPN [23] | 37.1 | 34.3 | 36.7 | 35.1 | **37.2** |
| Mask R-CNN [19] | 38.9 | 35.9 | 38.6 | 36.5 | **39.1** |
| PANet with DCN [27, 8] | 43.7 | 40.6 | 43.2 | 41.5 | **43.8** |

Table 2: Ablation study of network structures ($+$ indicates adding the module to the basic RNN).

We show the performance of our entire model and ablation study in Table 2. NMS is the ground-truth label for our network and RNN means using basic RNN module for encoder and decoder, which is the baseline. It is noticeable that using RNN module cannot produce reasonable results because summarizing all proposal candidates only according to hidden states is difficult. With global attention and context gate, the performance ameliorates. The reason that our final model performs best is that global attention can capture the relation for all proposal candidates. Context gate makes our model memorize the low-grade feature in high layers while the loss function is only based on the output score of our network rather than the origin score from detection network.

### 5.2 Stage II Evaluation

We take proposals selected by stage I with prediction score higher than $0.01$ as input. Weight of positive samples for our BCE loss is set to $2$.

| Model | NMS | Box Voting | Soft NMS | Stage I | Stage II (joint) | Stage II (step-by-step) |
|---|---|---|---|---|---|---|
| FPN [23] | 37.1 | 37.5 | 37.8 | 37.2 | 38.1 | **38.3** |
| Mask R-CNN [19] | 38.9 | 39.3 | 39.6 | 39.1 | 40.0 | **40.2** |
| PANet with DCN [27, 8] | 43.7 | 44.2 | 44.3 | 43.8 | 44.4 | **44.6** |

Table 3: Comparison of our approach and other alternatives. For NMS and Soft-NMS, we both use the best parameter $0.6$. We include global attention and context gate in each stage of our approach. Two training strategies are adopted respectively for comparison.

The performance of our model and prior solutions are compared in Table 3. With our full structure, the proposed method outperforms other popular duplicate removal solutions, including NMS, Soft-NMS and box voting.

For FPN and Mask R-CNN, our model trained with single head corresponding to IoU threshold $(0.5)$ increases more than one point and $0.9$ point even for the strong baseline, PANet with DCN, which generates more discriminative proposals.

| NMS | ours | sequence order | rank $f_R$ | appearance $f_A$ | box $f_G$ | origin score $f_S$ |
|---|---|---|---|---|---|---|
| | all | none | none | none | none | none |
| 37.1 | **38.3** | 33.8 | 37.6 | 35.4 | 37.5 | 36.9 |

Table 4: Ablation study of input features for our model (*none* indicates no such feature or out of order for the sequence, *all* means all input features in a descending order are used).

Ablation studies on the source of features are performed. The results are shown in Table 4. The order of sequence and appearance feature $f_A$ play important roles. Rank feature $f_R$, geometric feature $f_G$ and score feature $f_S$ help our model make prediction from more global view compared with NMS.

We analyze the importance of sample distribution. As shown in Table 5, we train stage II directly with output from detection network. Compared with our full framework, the performance drops severely. This manifests the necessity of conducting stage I to suppress easy negatives. We also take the result of NMS as input to stage II, however the mAP is slightly lower than using output of stage I. This comparison also shows that our structure is compatible with the box voting method.

| Model | FPN [23] | Mask R-CNN [19] | PANet [27] with DCN [8] |
|---|---|---|---|
| Detection Network | 33.8 | 35.5 | 40.1 |
| NMS | 38.1 | 40.0 | 44.4 |
| Stage I | **38.3** | **40.2** | **44.6** |

Table 5: Ablation study of the influence of input distribution on stage II. We directly take the output of detection network, NMS or stage I as the input to our second stage respectively.

| Model | FPN [23] | Mask R-CNN [19] | PANet [27] with DCN [8] |
|---|---|---|---|
| Stage II $(0.5)$ | 38.3 | 40.2 | 44.6 |
| Stage II $(0.75)$ | 38.4 | 40.3 | 44.5 |
| Stage II $(0.5 - 0.1 - 0.9)$ | **38.6** | 40.5 | **44.8** |
| Stage II $(0.5 - 0.05 - 0.9)$ | 38.6 | **40.6** | 44.8 |

Table 6: Comparison of using different IoU thresholds in the second stage. Last two rows use multiple thresholds with different intervals such as $0.1$ or $0.05$.

Table 6 compares the performance of utilizing different IoU thresholds when assigning the ground-truth labels at stage II. With multi-heads trained on samples assigned with multiple thresholds, our model further improves the performance by $0.3$, accomplishing a new state-of-the-art result.

We summarize our approach on *val* and *test-dev* subsets for different detection backbones trained with multiple thresholds in Table 7. We achieve nearly $1.5$ point improvement based on output from FPN and Mask R-CNN. With stronger baseline PANet with DCN, we also surpass the traditional NMS by $1.1$ points. It is noted that our model get larger improvement in $\text{mAP}_{75}$ than that in $\text{mAP}_{50}$, manifesting that our model makes good use of quality of proposals. The improvement statistics on COCO *test-dev* is similar.

| backbone | test set | mAP | $\text{mAP}_{50}$ | $\text{mAP}_{75}$ |
|---|---|---|---|---|
| FPN [23] | val | $37.1 \rightarrow 37.2 \rightarrow \textbf{38.6}$ | $59.0 \rightarrow 58.6 \rightarrow \textbf{59.6}$ | $39.8 \rightarrow 40.3 \rightarrow \textbf{42.3}$ |
| | testdev | $36.9 \rightarrow 37.0 \rightarrow \textbf{38.4}$ | $58.4 \rightarrow 58.1 \rightarrow \textbf{59.0}$ | $39.8 \rightarrow 40.3 \rightarrow \textbf{42.3}$ |
| Mask R-CNN [19] | val | $38.9 \rightarrow 39.1 \rightarrow \textbf{40.6}$ | $59.7 \rightarrow 59.4 \rightarrow \textbf{60.6}$ | $42.4 \rightarrow 43.1 \rightarrow \textbf{44.9}$ |
| | testdev | $39.1 \rightarrow 39.2 \rightarrow \textbf{40.6}$ | $59.6 \rightarrow 59.3 \rightarrow \textbf{60.1}$ | $42.7 \rightarrow 43.2 \rightarrow \textbf{45.1}$ |
| PANet with DCN[27, 8] | val | $43.7 \rightarrow 43.8 \rightarrow \textbf{44.8}$ | $63.4 \rightarrow 62.8 \rightarrow \textbf{63.4}$ | $47.9 \rightarrow 48.5 \rightarrow \textbf{49.6}$ |
| | testdev | $43.4 \rightarrow 43.4 \rightarrow \textbf{44.4}$ | $\textbf{63.0} \rightarrow 62.1 \rightarrow 62.4$ | $47.7 \rightarrow 48.1 \rightarrow \textbf{49.5}$ |

Table 7: Improvement from NMS to stage I and II (connected by $\rightarrow$ from left to right) based on different stage-of-the-art object detection systems on COCO2017 *val* and *test-dev*.

Fig. 3 shows that our approach reduces proposal candidates and increases performance at the same time. With output from FPN, Soft NMS keeps about $320.98$ proposals in one image on average, while our approach only produces $84.02$ proposals compared with $145.76$ from NMS using the same score threshold $0.01$.

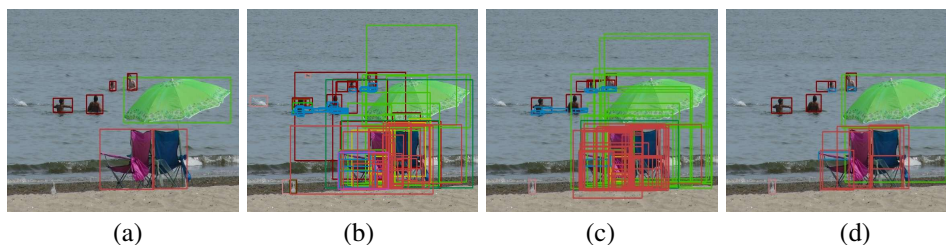

|     |     |     |     |
|---|---|---|---|
| (a) | (b) | (c) | (d) |

Figure 3: Visualization of ground truth (a), NMS (b), stages I (c) and II (d) of our approach.

## 6    Conclusion

We have presented a new approach for duplicate removal that is important in object detection. We applied RNN with global attention and context gate structure to sequentially encode context information existing in all object proposals. The decoder selects appropriate proposals as final output. Extensive experiments and ablation studies were conducted and the consistent improvement manifests the effectiveness of our approach. We plan to connect our framework to object detection networks to enable joint training for even better performance in future work.

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
