[Reviews · NeurIPS 2018]

Reviewer 1



- Abstract Although Duplicate Removal is an important step in object detection, since only the score of each candidate region is considered, the information of the input data was not sufficiently used by the conventional method. This paper proposes a new Duplicate Removal method based on RNN. Based on each candidate area, informative features are extracted by using appearance feature, position and ranking information in addition to the score. Then, they are treated as series data and are input into the RNN-based model to improve the final accuracy by capturing global information. The number of candidate regions is enormous to the number of objects that are to be left. Therefore, this paper proposes to reduce the box gradually by dividing it into two stages. In the two stages, the RNN model of the same structure was used. In stage I, to remove simple boxes the model is trained by using NMS results as a teaching signal. In stage II, to remove difficult boxes, the model is trained by using the grand-truth boxes. Experiments showed that mAP is increased in the SOTA object detection methods (FPN, Mask R - CNN, PANet with DCN) with the proposed method. - Pros: -- This paper is easy to understand. -- This paper pointed out the existing Duplicate Removal and proposed a new method to solve it. -- The proposed method is well designed. - Cons: -- The proposed method is complicated. -- Improvement of detection accuracy seems to be marginal against the complexity of the method. -- This paper does not include theoretical insights. -- Calculation of NMS is necessary for learning in stage I. -- There is no description about the calculation cost. Since object detection is a trade-off between detection accuracy and calculation speed mainly in inference, a description on calculation cost is indispensable. -- It would be better to try on other datasets. In addition, not only the overall mAP, but also more detailed analysis of the results is required. -- - In Table 7, the paper compares only with NMS. This paper should also compare with Box voting and Soft NMS. -- The publication year of [21] might be CVPR 2018. Although methods are well designed, it is unclear about the calculation cost, and theoretical insights and new findings are not enough at this time.

Reviewer 2



The work focuses o duplicate removal in object detection frameworks. It proposes a two-stage framework to effectively select candidates for objects: first easy negative objects proposals are eliminated and then true positives are selected from the reduced set of candidates. The two stages are based on an encoder and a decoder implemented as a recurrent neural network that uses global contextual information. The authors carried out extensive experiments showing how the approach outperforms other proposals on a benchmark dataset. The paper is well written and technically sound to the extent I checked (this is not my area of expertise). I liked the idea of a two-stage process in order to incorporate contextual knowledge and being more efficiently in the selection. Particular comments: - In section 3 there is no reference to the data set used for the preliminary empirical study. Is it the same as used later for performance? - "This is because that within any single" --> remove "that"

Reviewer 3



The paper describes an approach to learn to perform duplicate removal for object detection -- finding the right proposals for region classification. The proposed idea consists of two stages: in the first stage it learns to behave like NMS and reject most of the negative proposals; in the second stage it receives positive candidates from the first stage, and then select the final set of proposal outputs. Experiments are done on COCO with state of the art object detectors like FPN. - Overall I think it is a decent paper. Replacing NMS has been an interesting topic in the recent years for object detection so the problem is of significance. While the current approach is based on the previous work that use relation networks for object detection, it has introduced new components (two-stage, global attention etc). The paper is quite well written and well illustrated with figures. The experiments are extensive, with multiple state of the art detectors and multiple analysis. - It is missing citations for a recent work that also partially learns NMS: Chen, Xinlei, and Abhinav Gupta. "Spatial memory for context reasoning in object detection." ICCV (2017). - What is the training/validation set performance for stage I training? It is interesting that on the test set the learned model can "outperform" the teacher used for training. Is the 0.1-0.2 improvement statistically significant? I think the reason may be beyond what is mentioned in the paper. - Is stage I and stage II trained jointly or sequentially? - It would be interesting to see the total speed/memory usage for the proposed method. The paper mentioned that the proposed approach reduced the number of proposals used for second stage classification, however RoI operations can be done in parallel for multiple regions, whereas the entire region classification step will have to wait till the proposals are computed in the RPN. So I am not totally convinced about the speed argument.